# Modelling filovirus maintenance in nature by experimental transmission of Marburg virus between Egyptian rousette bats

Amy J. Schuh[1], Brian R. Amman[1], Megan E.B. Jones[1,2,†], Tara K. Sealy[1], Luke S. Uebelhoer[1,†], Jessica R. Spengler[1], Brock E. Martin[1,†], Jo Ann D. Coleman-McCray[1], Stuart T. Nichol[1] & Jonathan S. Towner[1,2]

The Egyptian rousette bat (ERB) is a natural reservoir host for Marburg virus (MARV); however, the mechanisms by which MARV is transmitted bat-to-bat and to other animals are unclear. Here we co-house MARV-inoculated donor ERBs with naive contact ERBs. MARV shedding is detected in oral, rectal and urine specimens from inoculated bats from 5–19 days post infection. Simultaneously, MARV is detected in oral specimens from contact bats, indicating oral exposure to the virus. In the late study phase, we provide evidence that MARV can be horizontally transmitted from inoculated to contact ERBs by finding MARV RNA in blood and oral specimens from contact bats, followed by MARV IgG antibodies in these same bats. This study demonstrates that MARV can be horizontally transmitted from inoculated to contact ERBs, thereby providing a model for filovirus maintenance in its natural reservoir host and a potential mechanism for virus spillover to other animals.

[1] Viral Special Pathogens Branch, Division of High-Consequence Pathogens and Pathology, Centers for Disease Control and Prevention, Atlanta, Georgia 30333, USA. [2] Department of Pathology, College of Veterinary Medicine, University of Georgia, Athens, Georgia 30602, USA. † Present address(es): Wildlife Diseases Laboratory, Institute for Conservation Research, San Diego Zoo Global, Escondido, California 92027, USA (M.E.B.J); Childhood Development and Rehabilitation Center, Oregon Health and Science University, Portland, Oregon 97239, USA (L.S.U.); Poxvirus and Rabies Branch, Division of High-Consequence Pathogens and Pathology, Centers for Disease Control and Prevention, Atlanta, Georgia 30333, USA (B.E.M.). Correspondence and requests for materials should be addressed to J.S.T. (email: jit8@cdc.gov).

Marburg virus (MARV; family *Filoviridae*, genus *Marbugvirus*), like its close relative Ebola virus, causes outbreaks of a rapidly progressive haemorrhagic disease perpetuated by human-to-human transmission, with case fatality ratios reaching up to 90% (ref. 1). The first recorded outbreak of MARV disease occurred in 1967 among former-West German and -Yugoslavian laboratory workers who contracted the disease while working with infected non-human primates imported from areas in Uganda where fruit bats were prevalent[2,3]. In the following decades, outbreaks of MARV disease occurred sporadically in sub-Saharan Africa and cumulative evidence suggested bats were involved[2–8]. Between 2007 and 2008, two outbreaks of MARV disease occurred in Southwest Uganda—one among miners working in Kitaka Mine[9] and the other in two tourists that had separately visited Python Cave in Queen Elizabeth National Park[10,11]. Follow-up ecological investigations revealed that both of these sites were inhabited by large populations of the cave-dwelling Egyptian rousette bat (ERB; *Rousettus aegyptiacus*)[9]. Longitudinal studies later identified this bat species as a natural reservoir host for MARV and a source of virus spillover into human populations[9,12]. This discovery was based upon several pieces of evidence including: (1) the consistent detection of both active and past infection in wild ERBs inhabiting caves or mines near MARV disease outbreaks, (2) a high genetic similarity between MARV sequences derived from wild ERBs and human outbreak isolates and (3) a temporal association between MARV disease spillover events, seasonal pulses of active MARV infection in juvenile ERBs and the biannual ERB-birthing seasons[9,12].

Experimental studies have focused on understanding aspects of the natural history of MARV infection in ERBs[13–15]. The first study found that ERBs inoculated by the intraperitoneal and subcutaneous routes with the Vero cell-adapted, human-derived Hogan strain of MARV exhibited viral replication in multiple tissues in the absence of clinical disease followed by seroconversion, while ERBs inoculated by the oronasal route (dripped virus into each nostril and on the tongue) with the same virus showed no evidence of MARV infection within the 21-day specimen collection period[13]. A second study using the low-passage, bat-derived 371 bat strain of MARV inoculated by the subcutaneous route demonstrated that infectious MARV was shed in oral secretions of experimentally infected ERB, indicating that the virus had potential to be horizontally transmitted between ERBs by the direct or indirect contact routes[14]. A more recent study was unable to show horizontal MARV transmission via direct, indirect or airborne routes from experimentally infected ERBs inoculated by the subcutaneous route with a low-passage, human-derived MARV strain (SPU 148/99/1) to in-contact naïve ERBs[15]. However, the experiment lasted only 42 days and serial euthanasia of experimentally infected ERBs likely resulted in fewer numbers of viral shedding bats, thereby reducing the probability of transmission to susceptible in-contact bats. Furthermore, the experimentally infected ERBs in this study shed little to no MARV in oral, nasal, penile, rectal and urine specimens. From these data, the authors suggested that bat-to-bat MARV transmission is mediated by haematophagous arthropods present in natural roosts.

In this study, we assess the potential for horizontal bat-to-bat transmission of MARV by co-housing 12 MARV-inoculated donor ERBs with 24 naïve contact ERBs and monitoring all bats for evidence of viral infection for 9 months. All bats are captive bred animals free of haematophagous arthropods. Herein, we demonstrate simultaneous MARV shedding from inoculated bats with oral exposure events in contact bats during the early study phase. After a prolonged period, viremia, shedding and seroconversion are detected in contact bats during the late study phase.

## Results

**General observations.** Groups of inoculated and contact ERBs were co-housed in various configurations to determine the potential for horizontal transmission of MARV in a controlled laboratory environment (Fig. 1). MARV RNA loads in biological specimens were measured by Q-RT–PCR and are reported hereafter as $\log_{10}$ 50% tissue culture infectious dose equivalents ml$^{-1}$ ($\log_{10}$TCID$_{50}$eq ml$^{-1}$). At 0 days post infection (DPI), none (0/38) of the bats had detectable viremias or MARV IgG antibodies (Fig. 2), indicating no prior exposure to MARV. Biological specimens obtained from the two negative control bats over the course of the study tested uniformly negative for MARV RNA and MARV IgG antibodies. Throughout the study, all bats appeared clinically healthy, and exhibited normal body weights and rectal temperatures. No parameter measured in this study (that is, MARV RNA loads, duration of MARV shedding and peak MARV IgG antibody levels) differed significantly between male and female bats (Supplementary Table 1), consistent with previous reports[9,12,16].

**Inoculated bats shed MARV during the early study phase.** All of the inoculated bats (12/12) developed MARV viremia (Fig. 3a). Peak viremias ranged from 1.4 to 3.0 $\log_{10}$TCID$_{50}$eq ml$^{-1}$ (mean peak load = 2.5 $\log_{10}$TCID$_{50}$eq ml$^{-1}$) and occurred between 5 and 12 DPI (mean day of mean peak load = 6.8 DPI). The interval of detectable viremia ranged from 1 to 16 DPI, with a mean duration of 6.0 days.

Oral MARV shedding was detected in 91.7% (11/12) of the inoculated bats (Fig. 3b). Peak oral shedding loads ranged from 1.6 to 5.4 $\log_{10}$TCID$_{50}$eq ml$^{-1}$ (mean peak load = 4.7 $\log_{10}$TCID$_{50}$eq ml$^{-1}$) and occurred between 6 and 14 DPI (mean day of mean peak load = 9.1 DPI). The interval of detectable oral shedding ranged from 5 to 19 DPI, with a mean duration of 4.6 days. The mean oral swab loads were higher than the mean blood loads from 6 to 19 DPI. Oral MARV loads exceeded the MARV inoculum dose of 4 $\log_{10}$TCID$_{50}$ in 33.3% (4/12) of the bats, consistent with virus replication in the bat. Infectious MARV was isolated from 17.6% (9/51) of the MARV Q-RT–PCR-positive oral swabs, confirming the shedding of infectious virus. MARV RNA loads were significantly higher (Mann–Whitney $U$-statistic = 93.0, $P = 0.0157$) in MARV isolation-positive oral swabs (mean = 4.5 $\log_{10}$TCID$_{50}$eq ml$^{-1}$, s.d. = 4.9) compared with MARV isolation-negative swabs (mean = 3.9 $\log_{10}$TCID$_{50}$eq ml$^{-1}$, s.d. = 4.6). The DPI of oral swab collection was significantly earlier (Mann–Whitney $U$-statistic = 72.5, $P = 0.0027$) in MARV isolation-positive oral swabs (mean = 7.8 DPI, s.d. = 2.3) compared with MARV isolation-negative swabs (mean = 11.3 DPI, s.d. = 3.3).

Rectal MARV shedding was detected in 33.3% (4/12) of the inoculated bats (Fig. 3c). Peak rectal shedding loads ranged from 0.7 to 1.2 $\log_{10}$TCID$_{50}$eq ml$^{-1}$ (mean peak load = 1.0 $\log_{10}$TCID$_{50}$eq ml$^{-1}$) and occurred between 6 and 8 DPI (mean day of mean peak load = 6.8 DPI). The interval of detectable rectal shedding ranged from 6 to 13 DPI, with a mean duration of 1.5 days. The mean rectal swab loads were lower than the mean blood loads (Fig. 3c), indicating that contribution of signal from contaminating blood could not be ruled out. The internal extraction control failed to be detected in 6.1% (22/360) of the rectal swab specimens taken from inoculated bats, indicating the presence of inhibitory substances that may have interfered with RNA extraction. Attempts to dilute-out the inhibitory substances were unsuccessful.

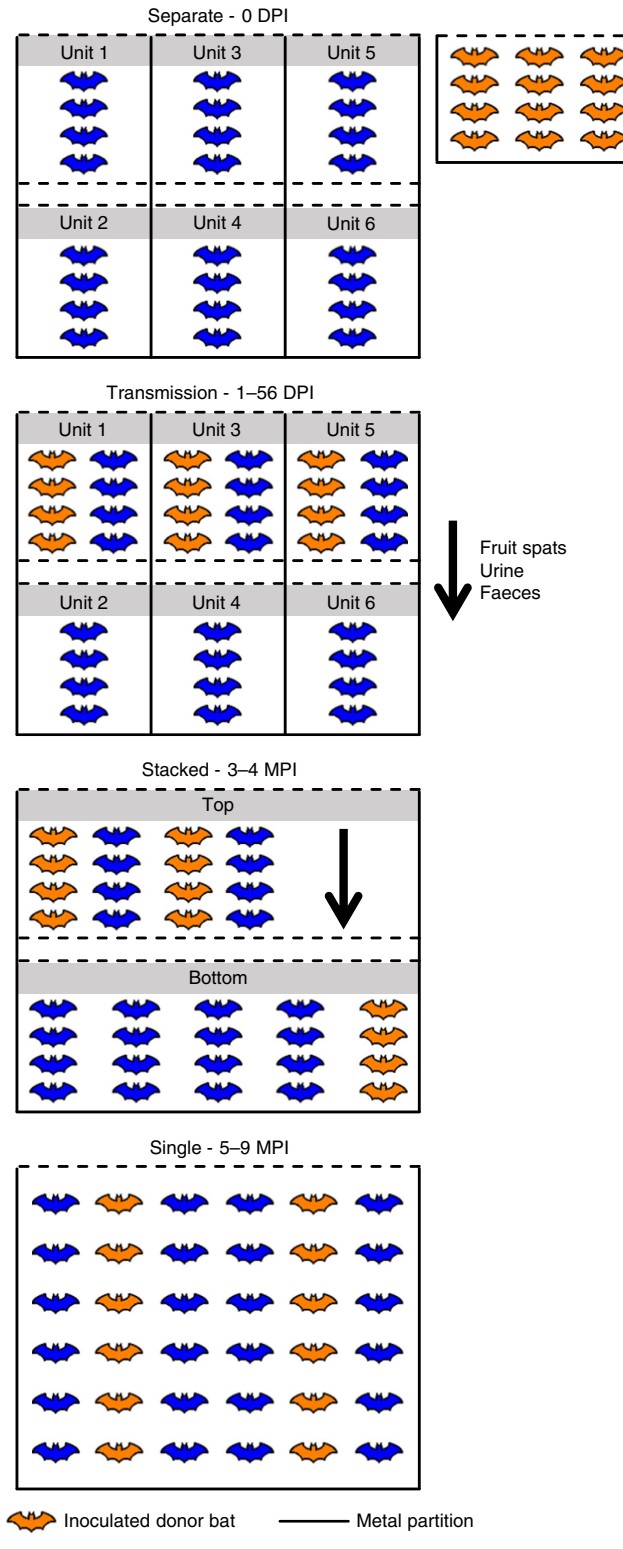

**Figure 1 | Housing configurations throughout the study.** Bats were transferred to different housing configurations at the time indicated above each caging diagram.

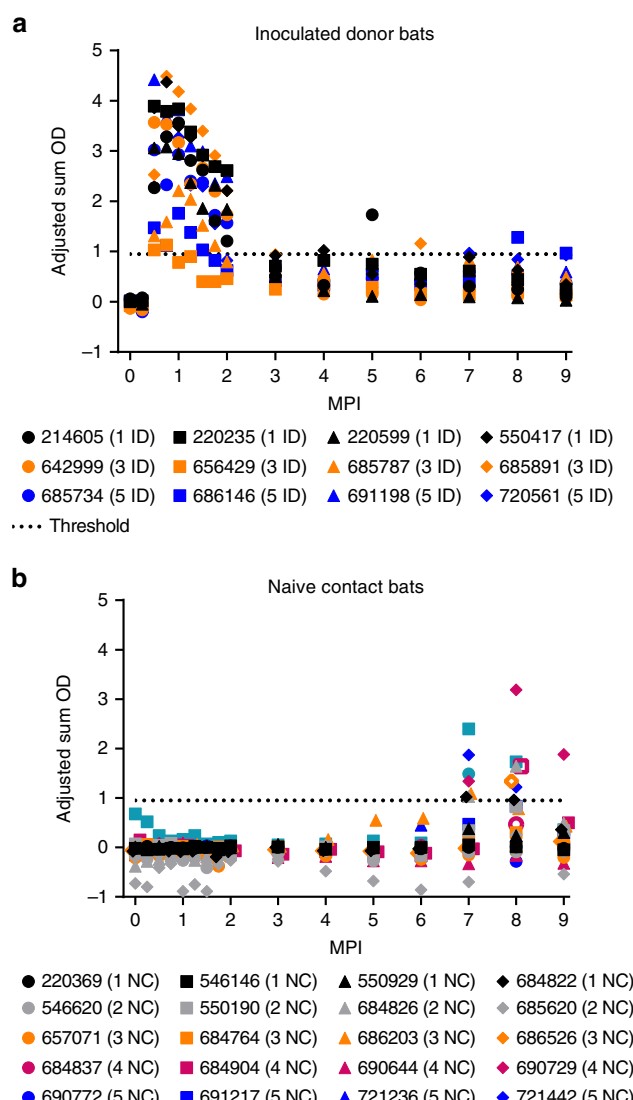

**Figure 2 | MARV IgG antibody responses in inoculated donor and naive contact bats.** MARV IgG antibody responses in (**a**) inoculated and (**b**) contact bats, as detected by ELISA with purified recombinant nucleoprotein of the Angola strain of MARV expressed in *Escherichia coli*. The dashed lines represent the cutoff value of the assay (MARV seropositive $\geq 0.95$). The legends indicate the housing unit (1–6) and group (inoculated donor-ID and naive contact-NC) of each bat during the early study phase. Open symbols in **b** at 8 MPI represent the three contact bats which were viremic at 7 MPI.

MARV was detected in the urine of 16.7% (2/12) of the inoculated bats. Of these bats, one had five positive specimens (range of detection = 10–16 DPI, peak load = 3.3 $\log_{10}\text{TCID}_{50}\text{eq ml}^{-1}$ at 10 DPI) and the other had a single positive specimen (16 DPI, 0.5 $\log_{10}\text{TCID}_{50}\text{eq ml}^{-1}$; Fig. 3d). It is likely that a higher proportion of inoculated bats shed virus in their urine, as only 28.1% (101/360) of the attempts to collect urine were successful and many of these collections fell outside the expected range of viral shedding (that is, 5–15 DPI; Supplementary Table 2). The mean virus genome equivalents of 66.7% (4/6) of the MARV-positive urine specimens were higher than the mean virus genome equivalents of the blood specimens. Infectious MARV was not isolated from the only urine specimen in which the specimen quantity was sufficient to attempt virus isolation. Notably, both bats that had detectable MARV RNA in their urine also had positive oral and rectal swabs.

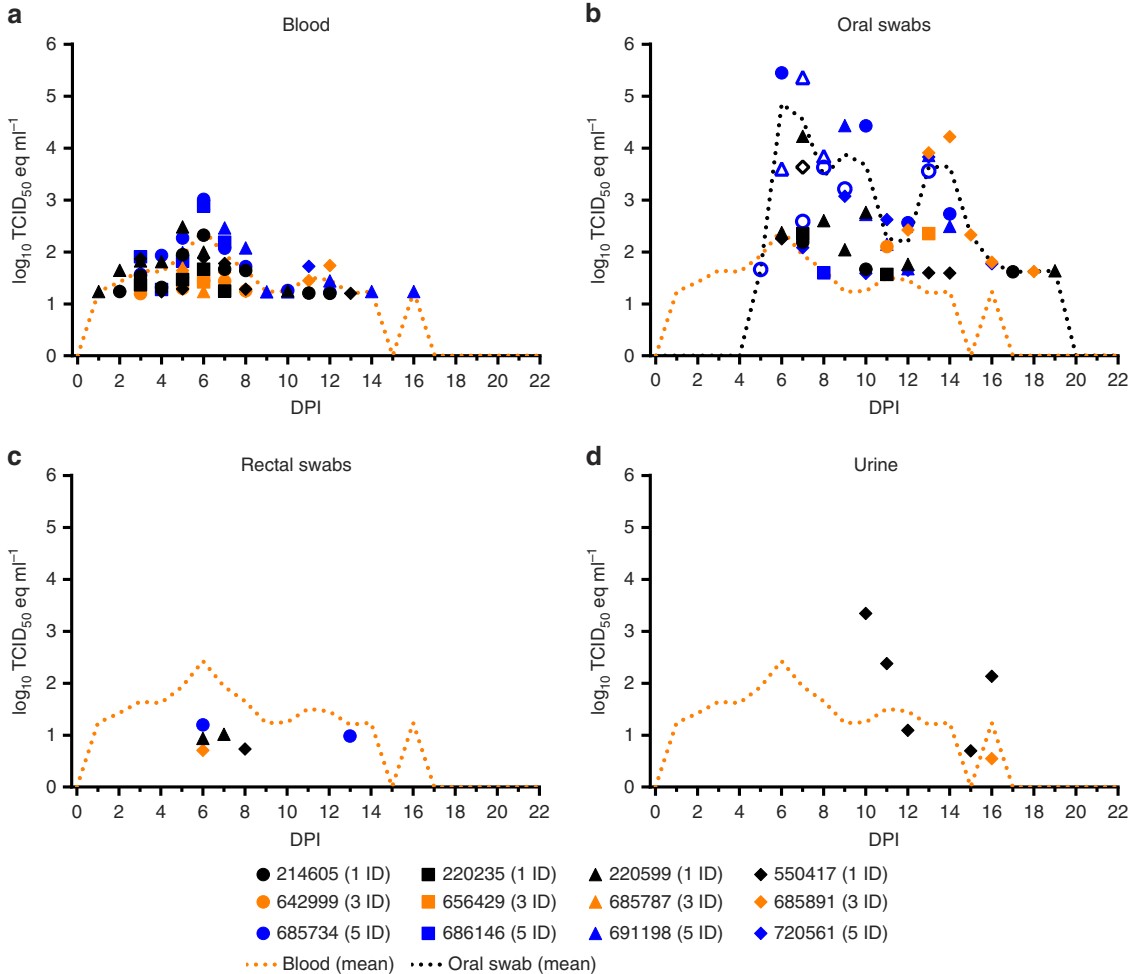

**Figure 3 | MARV loads in specimens obtained from inoculated donor bats.** MARV loads (Q-RT–PCR-derived $\log_{10}TCID_{50}eq\,ml^{-1}$) in (**a**) blood, (**b**) oral swabs, (**c**) rectal swabs and (**d**) urine. The orange dotted lines in **a**–**d** represent the overall mean MARV load in blood specimens and the black dotted line in **b** represents the overall mean viral load in oral swabs. The legend indicates the housing unit (1, 3 or 5) and group (inoculated donor-ID) of each bat during the early study phase. Open symbols in **b** represent oral swabs from which infectious MARV was isolated. The lower limit of detection was 0.4, 0.7, $-0.3$ and $-0.3$ $\log_{10}TCID_{50}eq\,ml^{-1}$ for blood, oral swabs, rectal swabs and urine, respectively.

All of the inoculated bats (12/12) seroconverted to MARV (mean peak adjusted sum optical density (OD) = 3.23, s.d. = 1.07), with MARV IgG antibodies peaking between 14 and 28 DPI (day of mean peak MARV IgG antibodies = 20.4 DPI) and then declining (Fig. 2a). Peak levels of MARV IgG antibodies coincided with the end of detectable viral shedding (Figs 2 and 3b–d). Four inoculated bats exhibited marked increases in MARV IgG antibodies between 5 and 8 months post infection (MPI)—bat 214605 (trough sum OD = 0.33 at 4 MPI, peak sum OD = 1.73 at 5 MPI), bat 685891 (trough sum OD = 0.29 at 5 MPI, peak sum OD = 1.16 at 6 MPI), bat 720561 (trough sum OD = 0.33 at 4 MPI, peak sum OD = 0.97 at 7 MPI) and bat 686146 (trough sum OD = 0.33 at 4 MPI, peak sum OD = 1.28 at 8 MPI).

**Individual heterogeneities in oral MARV shedding.** Viral shedding is a measure of infectiousness[17] and was calculated for each inoculated bat by summing MARV RNA loads detected over time in oral swabs during the early study phase (0–56 DPI). MARV loads detected in rectal swab and urine specimens were not included in this calculation, as these specimens were subjected to specimen integrity bias and collection bias, respectively. Total MARV oral shedding varied markedly among the 12 inoculated bats, with sum $\log_{10}TCID_{50}eq\,ml^{-1}$ ranging from undetectable to 5.5 (mean = 4.7, s.d. = 5.1; Fig. 4a). The Lorenz curve and associated Gini coefficient (0.80) further demonstrate individual heterogeneity in MARV oral shedding and show that a minority of the inoculated bat population was responsible for a disproportionately large percentage of viral shedding (Fig. 4b). For example, the curve shows that 25.0% of the inoculated bat population was responsible for 98.3% of MARV oral shedding, 50.0% of the bats were responsible for 99.4% of oral shedding and 75.0% of the bats were responsible for 99.4% of oral shedding. Using a previously established approach[18], two inoculated bats were classified as supershedders (685734 and 691198), as each of them shed virus at levels greater than the 80th percentile (4.82 $\log_{10}TCID_{50}eq\,ml^{-1}$) and together accounted for 91.1% of the total MARV oral shedding. MARV oral shedding was detected in both of these bats at nine time points each, and infectious virus was isolated from five oral swabs taken from bat 685734 and three oral swabs taken from bat 691198.

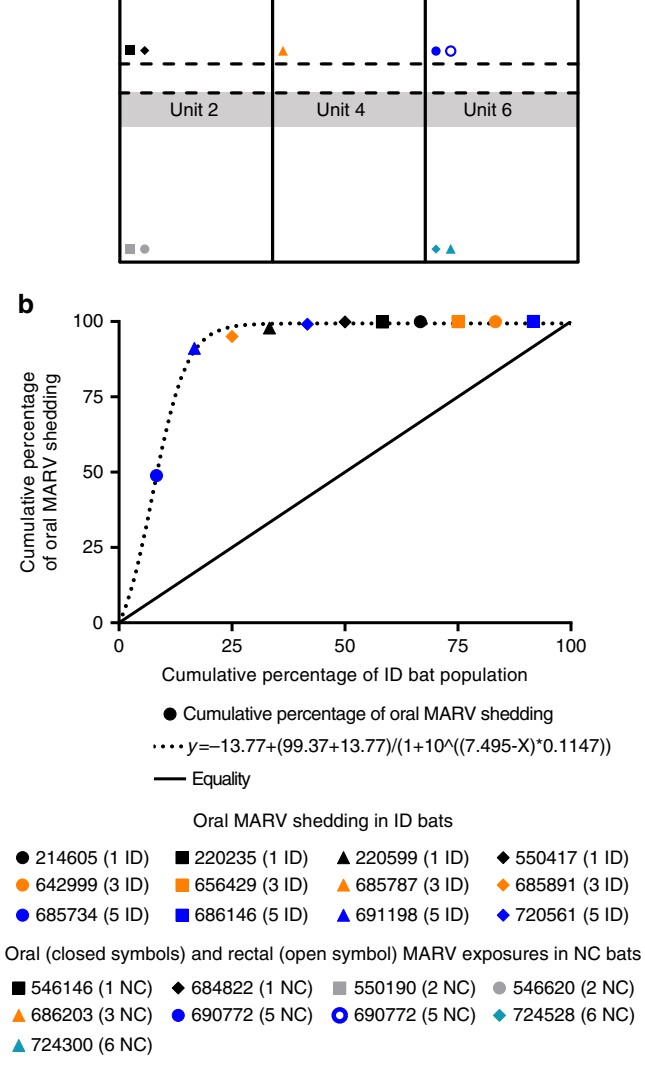

**Figure 4 | Cumulative oral MARV shedding from inoculated bats and contact bat exposures.** (**a**) Cumulative oral MARV shedding in $\log_{10}\text{TCID}_{50}\text{eq ml}^{-1}$ from each of the inoculated bats (symbols at the top of each unit) and the number of exposures in the contact bats (symbols at the bottom of each unit; exposure defined as a Q-RT–PCR-positive oral or rectal swab) according to unit and bat, and (**b**) Lorenz curve of cumulative percentage of the inoculated bat population versus cumulative percentage of oral MARV shedding ranked in descending order (that is, first circle on the curve represents bat 685734, which had the highest cumulative percentage of oral MARV shedding). The legends indicate the housing unit (1–6) and group (inoculated donor-ID and naive contact-NC) of each bat during the early study phase.

**Contact bats exposed to MARV during the early study phase.** During the early study phase (0–56 DPI), none of the contact bats (0/24) had detectable viremias. However, MARV RNA was detected in oral swabs from eight contact bats, at one time point each, between 4 and 22 DPI and one contact bat had a MARV-positive rectal swab at 9 DPI (Fig. 5). Infectious MARV was not isolated from any of the eight MARV Q-RT–PCR-positive oral swabs. None of the contact bats (0/24) seroconverted

during the early study phase (0–56 DPI; Fig. 2b). The detection of MARV RNA in oral and rectal swabs from contact bats within 24 h of detecting virus in inoculated bats, combined with the lack of seroconversion in contact bats, indicates that the contact bats were exposed to the virus during the early study phase through activities such as consuming virus-contaminated food or water and social grooming.

A total of 4.4 $\log_{10}\text{TCID}_{50}\text{eq ml}^{-1}$ of MARV shed by inoculated bats in unit 1 resulted in two exposures in the same unit and two exposures in the unit below (unit 2; Fig. 4a). A total of 4.4 $\log_{10}\text{TCID}_{50}\text{eq ml}^{-1}$ of MARV shed by inoculated bats in unit 3 resulted in one exposure in the same unit and zero exposures in the unit below (unit 4). A total of 5.8 $\log_{10}\text{TCID}_{50}\text{eq ml}^{-1}$ of MARV shed by inoculated bats in unit 5 resulted in two exposures in the same unit and two exposures in the unit below (unit 6).

**MARV infection of contact bats during the late study phase.** After the early study phase (0–56 DPI), the contact and inoculated bats were transferred to gang housing in one of two stacked cages (3–4 MPI) or a single large cage (5–9 MPI; Fig. 1). During this time, bats were sampled on a monthly basis. At 7 MPI, MARV viremia was detected in three contact bats (684837, 684904 and 686526) and a positive oral swab was obtained from one of these bats (684904; Fig. 5). Two of these bats (684904 and 686526) seroconverted at 8 MPI, while the third bat exhibited a rise in MARV IgG antibody levels but failed to cross the threshold of seropositivity (684837; Fig. 2b).

By 9 MPI, 37.5% (9/24) of the contact bats had seroconverted (mean peak sum-adjusted OD = 1.74 with s.d. = 0.68; Fig. 2b). MARV IgG antibody levels in contact bat 686203 began to rise from the baseline level (sum OD = −0.03) at 5 MPI (sum OD = 0.55) and continued to increase through 6 MPI (sum OD = 0.59) until seroconversion occurred at 7 MPI (sum OD = 1.09; Fig. 2b), suggesting infection occurred around 4 MPI. Six additional contact bats seroconverted at 7 MPI, followed by two contact bats at 8 MPI and zero bats at 9 MPI. After seroconversion, IgG antibody titres declined. Peak MARV IgG antibody levels in contact bats were significantly lower ($n = 9$, mean peak sum-adjusted OD = 1.74 with s.d. = 0.68) than peak levels in inoculated bats ($n = 12$, mean peak adjusted sum OD = 3.23 with s.d. = 1.07; unpaired $t$-test statistic = 3.6, degree of freedom = 19, $P = 0.0017$). The seroconversions in contact bats coincided with the boosting effect observed in four inoculated bats between 5 and 8 MPI (detailed above) and indicates that MARV was circulating in the study population between 4 and 7 MPI.

**Discussion**
This study shows that MARV can be horizontally transmitted from infected ERBs to naive contact bats in the absence of arthropod vectors. Daily specimen collection during the early study phase showed that the majority (11/12) of inoculated bats shed MARV in their oral secretions, rectal excretions and/or urine between 5 and 19 DPI. Given the multiple routes and high levels of viral shedding observed in the inoculated bats, it was not surprising that low levels of MARV RNA were simultaneously detected in the oral mucosa of eight contact bats, housed in both the upper and lower units of the transmission cage. Notably, bats in the lower cages were not in direct contact with the inoculated donor bats. These findings not only indicate that MARV is horizontally transmitted bat-to-bat through direct and/or indirect contact with infectious bodily fluids[14], but also suggest that the virus may be transmitted to other animals, including humans, by the same mechanisms.

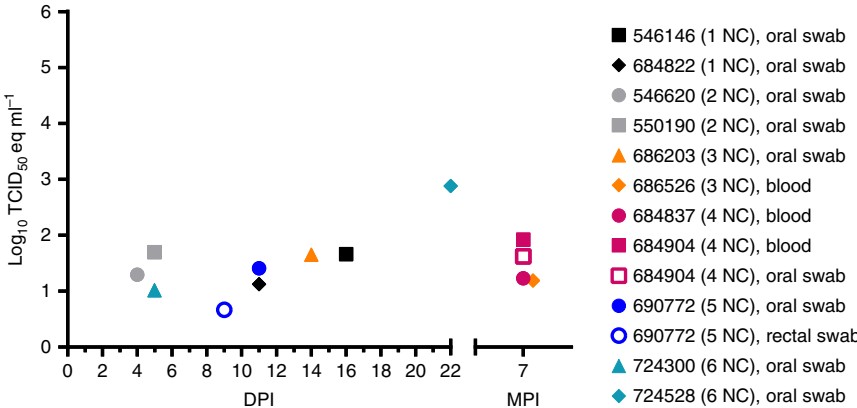

**Figure 5 | MARV loads in biological specimens obtained from naive contact bats.** MARV loads (Q-RT–PCR-derived $\log_{10} \text{TCID}_{50}\text{eq ml}^{-1}$) in oral swabs, rectal swabs and blood obtained during the early and late study phases (that is, 7 MPI). The legend indicates the housing unit (1–6) and group (naive contact-NC) of each bat during the early study phase. The lower limit of detection was 0.4, 0.7 and $-0.3 \log_{10}\text{TCID}_{50}\text{eq ml}^{-1}$ for blood, oral swabs and rectal swabs, respectively.

Plausible scenarios could be consumption of fruit or fruit spats by non-human primates that were contaminated by infectious saliva or urine, or for humans, introduction of infectious urine or faeces directly into a mucous membrane. This hypothesis is supported by MARV outbreak investigations revealing that many of the index cases had entered ERB-inhabited caves before becoming ill, but none had ever reported being bitten or scratched by a bat while in the cave[6,9–11]. Although the natural reservoir hosts of the other filoviruses have yet to be identified, mounting epidemiological data from outbreak investigations[19–23], ecological surveillance studies of filovirus in bats[24–33] and bat–filovirus experimental infections[34] point to bats as the most likely reservoir hosts and sources of virus spillover to humans. Therefore, this study provides: (1) an experimental model for how filoviruses are maintained in their natural bat reservoir host with occasional spillover to humans and other animals and (2) a plausible evidence-based mechanism for how the Ebola outbreak in West Africa may have started.

During the late study phase and coinciding with the boosting effect observed in inoculated bat 214605, an initial rise in MARV IgG antibody levels was observed in contact bat 686203 at 5 MPI, indicating that infection occurred around 4 MPI. After a relatively long lag-time this contact bat seroconverted at 7 MPI, along with an additional six contact bats. Moreover, at 7 MPI, MARV viremia was detected in three contact bats, and one of these bats also had a MARV RNA-positive oral swab. Two of these bats seroconverted at 8 MPI, while the third failed to seroconvert. Four inoculated bats exhibited marked increases in MARV IgG antibody levels between 5 and 8 MPI, providing further evidence that MARV was circulating in the study population between 4 and 7 MPI.

A recent 42-day investigation was unable to demonstrate horizontal MARV transmission via direct, indirect or airborne routes from inoculated ERBs experimentally infected with a human-derived MARV strain to naive ERBs[15]. Decreasing numbers of experimentally infected bats due to the serial euthanasia schedule in this study may have reduced the probability of infection of susceptible in-contact bats. The majority of experimentally infected bats had been euthanized by 9 DPI, a time point that coincided with the mean peak oral MARV shedding from the inoculated bats in our study. Furthermore, oral, nasal, vaginal, penile, rectal and urine specimens from experimentally infected ERBs in the previous investigation indicated that MARV shedding was undetectable in many inoculated bats and relatively low in others (range = 0.53–1.57 $\log_{10}$ $\text{TCID}_{50}\text{ml}^{-1}$). It is possible that the MARV exposure dose provided by the experimentally infected ERBs was insufficient to infect the in-contact bats because host-specific genetic mutations present in the human-derived MRV inoculum (SPU 148/99/1) led to the observed low-level viral shedding and decreased infectivity of the experimentally infected bats. Finally, the results of our study suggest that the study period in the previous investigation may have been too short to detect a transmission event through seroconversion of a contact bat.

Host populations exhibit marked heterogeneities in pathogen transmission, with a minority of infectious hosts producing the majority of secondary infections[18,35,36]. Transmission heterogeneity can result from interaction of the host with the pathogen and/or the interaction of the host with other hosts[17]. Given the same exposure dose, supershedders yield more organisms than other hosts, and this trait has been linked to intrinsic host factors, including genetic differences[37], immune suppression[38] and co-infections[39]. Superspreaders are individuals that have more opportunities to infect other hosts, typically through higher contact rates. Although we did not directly measure bat contact rates, daily oral swab collection throughout the infectious period of the inoculated bats allowed individual heterogeneities in oral MARV shedding to be examined. We found that 20.0% of the inoculated bats were responsible for the majority (95.4%) of oral MARV shedding. Two bats were classified as supershedders[18], as each of them generated greater than 20% of the total oral MARV shedding (bat 685734 = 48.8% and bat 691198 = 42.3%) and had total oral viral RNA shedding levels that exceeded the 80th percentile. Furthermore, MARV RNA was detected in oral swabs collected from both supershedder bats at nine time points, and infectious virus was isolated from these swabs eight times over nine days. Supershedding has been observed in multiple host–agent systems[18,40,41] and cohabitation with supershedders has been associated with increased levels of pathogen shedding from secondary infection cases[40]. Assuming that MARV shedding is directly proportional to infectiousness, supershedders may contribute substantially to the maintenance of MARV in natural ERB populations.

Our data indicate that following 'natural' laboratory infection with MARV (naive contact bats infected by contact with inoculated bats), ERBs may become viremic, shed virus

in their oral secretions and 1 month later develop peak MARV IgG antibody levels that rapidly decline within 1 month. These infection dynamics are consistent with those observed in the experimentally infected inoculated bats, where MARV IgG antibody levels typically peak 1 month following viremia and then rapidly decline. In nature, only 5% of the ERBs with MARV-positive liver/spleen pools also had MARV-positive blood specimens[12]. Past experimental infection of ERBs with MARV revealed that the virus is detectable in the liver and spleen of all bats through 8 DPI, but is detectable only in the blood of all bats when liver and spleen MARV loads peak at 5 and 6 DPI[14]. These data indicate that the three contact bats viremic at 7 MPI likely had high liver–spleen MARV loads and further suggest that naturally MARV-infected ERBs do not uniformly have detectable viremias. The mean peak MARV IgG antibody levels were 46% lower in the contact bats compared with the inoculated bats. Significantly decreased MARV IgG antibody levels following 'natural' infection is likely a function of inoculum dose and route, where 'naturally' infected contact bats may have been infected with a smaller inoculum dose and/or by a different route (that is, oral, ocular and/or intramuscular) than the inoculated bats (subcutaneously inoculated with a moderate dose of 4 $\log_{10}TCID_{50}$ of MARV). MARV IgG antibody levels in both inoculated and contact bats rapidly declined 1 month following the attainment of peak levels, and levels in inoculated bats dropped below the threshold of seropositivity by the following month. It is unclear whether waning MARV IgG antibody levels are indicative of diminishing protective immunity against viral reinfection, replication and shedding.

The MARV transmission chain (who infected who and when) could not be delineated in this investigation because of infrequent specimen collection and the transfer of bats to a single, large flight cage after the early study phase (0–56 DPI). At 5 MPI, we observed a boosting effect in inoculated bat 214605 and an initial rise in MARV IgG antibodies in contact bat 686203. This indicates that MARV was circulating in the study population just before this time and that contact bat 6860203 was infected around 4 MPI, if assuming a 21-day latent period. A mathematical model of MARV transmission in a closed ERB population found that a longer (21-day) latent period was necessary for virus maintenance, while a shorter period (7-day) resulted in MARV extinction[42]. The 3-month time gap between MARV shedding from the inoculated bats and infection of contact bat 686203 around 4 MPI can be explained by at least three scenarios. Given the rapid decline in MARV IgG antibodies 1 month following peak antibody levels in both inoculated and contact bats, it is possible that the monthly sampling schedule in the late study phase did not permit the serological detection of intermediary infectious contact bats linking the infectious inoculated bats to contact bat 686203. This was exemplified here, as contact bat 684837 was viremic at 7 MPI but failed to cross the threshold of seropositivity at 8 MPI. Others have noted rapidly waning virus-specific IgG[15,43] or viral neutralizing antibodies[44–46] following experimental inoculation of bats with a number of viruses including Marburg, Japanese encephalitis, Nipah, rabies, Hendra and European bat lyssavirus type 1. If the MARV–ERB relationship is reflective of the ebolavirus–bat relationship, this may explain why there is only scant serological evidence of Ebola virus infection in wild-caught bats[47]. A second possibility is that contact bat 686203 was exposed during the early study phase through contact with an infectious inoculated bat and after a long latent period of 3 months became infectious at 4 MPI. A 'natural' MARV infection with a lower-dose inoculum and/or by a different route may have led to a longer latent period compared with an experimental infection with a moderate-dose

inoculum by the subcutaneous route. Previous inoculation of ERBs by the oronasal route showed no evidence of MARV infection after 21 days[13]. Lower-dose rabies virus inoculums have been associated with longer incubation periods in experimentally infected Mexican free-tailed bats[48]. However, this scenario is not supported by a longitudinal investigation that showed active MARV infection can be found in newly susceptible 3-month-old juveniles[12], indicating that these individuals experienced a latent period no longer than 1 month. A third possibility is that one of the inoculated bats was persistently infected, intermittingly shed virus and infected contact bat 686203 at 4 MPI. Several bat-borne viruses, including Rio Bravo[49,50], Entebbe bat salivary gland[51], Montana myotis leukoencephalitis[52] and rabies viruses[48,53], have been detected in the saliva and/or salivary glands of wild-caught bats and experimentally infected bats long after virus infection. However, MARV RNA or infectious virus was not detected in the oral mucosa in this study after 19 DPI and in the salivary glands in previous studies after 14 DPI[13–15]. Further, MARV has been detected in reproductive tissue of a wild-caught ERB only once[12] and has never been detected in reproductive tissue of experimentally infected bats after 12 DPI[13–15].

This study opens the door for future investigations aimed at identifying the route of MARV horizontal transmission between ERBs (for example, direct contact with infectious bodily fluids, indirect contact with infectious bodily fluids and/or biting). Further experimental studies with MARV-infected ERBs are also needed to determine whether: (1) previously infected bats with low or undetectable IgG antibody levels can become infectious following a re-exposure to the virus, (2) a low-dose infection by either subcutaneous or oronasal routes results in an extended latent period and (3) bats become persistently infected and intermittently shed infectious virus. Answers to these questions are critical for obtaining a detailed understanding of how MARV is maintained in its natural reservoir host and spillover to humans occurs.

## Methods

**Animal procedures.** All animal procedures described herein were approved by the Centers for Disease Control and Prevention (CDC, Atlanta, Georgia, USA) Institutional Animal Care and Use Committee and were performed in accordance with the Guide for the Care and Use of Laboratory Animals (Committee for the Update of the Guide for the Care and Use of Laboratory Animals 2011). The CDC is an Association for Assessment and Accreditation of Laboratory Animal Care (AAA-LAC) fully accredited research facility.

Procedures conducted with MARV or MARV-infected bats were performed at the CDC under biosafety level 4 (BSL-4) laboratory conditions in compliance with Select Agent Regulations (Animal and Plant Health Inspection Service and Centers for Disease Control 2014). All investigators and animal caretakers followed BSL-4 biosafety and infection control practices to prevent cross-contamination between study groups. All bat cages were placed within bio-flow isolator units with HEPA-filtered inlet and exhaust air supplies (Duo-Flow Mobile Units, Lab Products Inc., Seaford, DE, USA).

All bats were housed in a climate-controlled BSL-4 animal area, with a 12 h day/12 h night cycle. Bats were provided daily with their body mass in fresh fruits supplemented with protein/vitamin powder (Lubee Bat Conservancy, Gainesville, FL, USA) and they received water ad libitum.

**Experimental design.** A total of 38 captive-born juvenile ERBs (6–7 m of age; 21 males and 17 females) were used in this study (Fig. 6). This MARV-free colony was founded from wild-caught ERBs imported from Uganda, as previously described[14]. Before importation, the quarantine housing of the ERBs was treated with permethrin, and the bats were initiated on a regimen of ivermectin and praziquantel.

ERBs were acclimated to the BSL-4 laboratory for 5 days before the beginning of the study (acclimation study phase) (Table 1). At 0 DPI, 12 inoculated donor bats were inoculated subcutaneously under isoflurane anaesthesia with 4 $\log_{10}TCID_{50}$ ($TCID_{50}$ = plaque forming units*7.33 (ref. 54)) of the 371 bat strain (Vero E6 + 2 passages; mycoplasma-free) of MARV[9] prepared in 0.25 ml of sterile DMEM (Life Technologies, Carlsbad, CA, USA) in the caudal abdominal region, two negative control bats were inoculated in the same manner as the inoculated bats

| Unit 1 | | Unit 3 | | Unit 5 | |
|---|---|---|---|---|---|
| 214605-M | 220369-M | 642999-M | 657071-M | 685734-M | 690772-M |
| 220235-M | 546146-M | 656429-M | 684674-M | 686146-M | 691217-M |
| 220599-F | 550929-F | 685891-F | 686203-F | 691198-F | 721236-F |
| 550417-F | 684822-F | 685787-F | 686526-M | 720561-M | 721442-F |

| Unit 2 | Unit 4 | Unit 6 |
|---|---|---|
| 546620-M | 684837-M | 720802-M |
| 550190-M | 684904-M | 724300-M |
| 684826-F | 690644-F | 724528-F |
| 685620-F | 690729-F | 726397-F |

**Figure 6 | Bat identification numbers according to group and unit.**
Inoculated donor bats are coloured orange and naive contact bats are
coloured blue.

## Table 1 | Study timeline.

| Time | Study phase | Housing | Specimen collection schedule | Specimen type(s) |
|---|---|---|---|---|
| − 5 to − 1 DPI | Acclimation | Random | None | None |
| 0 DPI | Inoculation | Separate | Daily | Blood |
| 1–25 DPI | Initial Transmission | Daily | B, O, R, U |
| 28–56 DPI | Initial Transmission | Weekly | B, O, R, U |
| 3–4 MPI | Late | Stacked | Monthly | Blood |
| 5–9 MPI | Late | Single | Monthly | Blood |

B, blood; DPI, days post infection; M, months post infection; O, oral swab; R, rectal swab;
U, urine.

with 0.25 ml of sterile DMEM and 24 naive contact bats received no inoculum (inoculation study phase; separate cages). Adhering to standard practice for viral transmission studies[55], the inoculated bats were housed separately from the contact bats through 1 DPI to prevent the possibility of transmission occurring between the two groups as a result of residual virus that may have been present at the inoculation site of the inoculated bats. At 1 DPI, the inoculated and contact bats were transferred to a transmission cage partitioned into six units, consisting of two rows and three columns maintained within a single bio-flow isolator unit (early study phase; transmission cage). The three columns were vertically separated by double solid metal partitions, while the two rows of caging were separated by a two-inch gap and wire caging only. Each of the three units within the top row contained four inoculated bats plus four contact bats and allowed for direct contact, while each of the three units within the bottom row contained four contact bats only. Bats in the top row could not come into direct contact with bats in the bottom row; however, fruit spats, urine and faeces could drop through wiring from a top unit into a bottom unit directly underneath. From 3–4 MPI, the inoculated and contact bats were co-housed within two separate compartments of a stacked cage maintained within a bio-flow isolator unit (late study phase; stacked cage), and from 5 to 9 MPI the inoculated and contact bats were co-housed in a single flight cage maintained within a bio-flow isolator unit (late study phase; single cage). Throughout the study, the two negative control bats were housed in a cage maintained within a separate bio-flow isolator unit. At 9 MPI, this study was concluded and all animals were transferred to another protocol.

**Monitoring and specimen collection.** Monitoring and specimen collection procedures have been previously described in full detail[14,16]. Rectal temperatures were taken daily from 0–25 DPI and then weekly through 56 DPI. Weights were recorded weekly from 0 to 56 DPI and then monthly through 9 MPI. Blood was taken before inoculation at 0 DPI (pre-bleed), and then daily through 25 DPI, weekly through 56 DPI and monthly through 9 MPI (Table 1). Oral, rectal and urine samples were obtained daily from 1 to 25 DPI and then weekly through 56 DPI. Oral swabs and urine were collected from select bats at 7 MPI.

Blood was taken from the cephalic vein using a sterile lancet (C&A Scientific, Manassas, VA, USA), two polyester-tipped applicators (Fisher Scientific, Grand Island, NY, USA) were used to swab the oral mucosa and a single rectal swab was obtained opportunistically at the time of rectal temperature by repurposing the plastic thermometer probe cover (MABIS Healthcare, Waukegan, IL, USA). Urine collection was attempted from the inoculated and negative control bats by allowing a single bat to hang in a sterile mouse cage fitted with an inverted wire top (Thoren Caging, Hazleton, PA, USA) and later collecting the accumulated urine.

## Table 2 | Specimen processing.

| Type | MARV Q-RT-PCR | MARV IgG antibodies* | MARV isolation† |
|---|---|---|---|
| Blood | X | X | |
| Oral swab | X | | X |
| Rectal swab | X | | |
| Urine‡ | X | | X |

MARV, Marburg virus; Q-RT-PCR, quantitative reverse transcriptase PCR.
*Once weekly.
†Only Q-RT-PCR-positive specimens.
‡Inoculated donor and negative control bats only.

Aliquots of whole blood were used to monitor for viremia (MARV RNA by Q-RT–PCR) and an antibody response (MARV-specific IgG antibody by ELISA; Table 2). Oral swabs, rectal swabs and urine were used to detect virus shedding and/or exposure (MARV RNA by Q-RT–PCR). In addition, oral swabs and urine (given sufficient quantity) were frozen in 0.5 ml of sterile DMEM to test for infectivity by virus isolation on Vero E6 cells.

**RNA extraction and Q-RT-PCR.** As previously described[14,16], RNA was extracted from gamma-irradiated-Rift Valley fever virus-spiked (RNA extraction-positive control) blood, oral, rectal and urine samples inactivated in lysis buffer solution using the MagMAX Pathogen RNA/DNA Kit (Life Technologies) with the MagMAX Express-96 Deep Well Magnetic Particle Processor (Life Technologies).

Reverse-transcribed MARV and Rift Valley fever virus RNA were detected on the ABI 7500 Real-Time PCR System (Life Sciences) using the SuperScript III Platinum One-Step Q-RT-PCR Kit (Life Technologies), with amplification primers and reporter probes targeting the viral protein 40 gene and the large segment, respectively (Supplementary Table 3). Relative MARV $TCID_{50}$eq ml$^{-1}$ were interpolated from standard curves generated from serial dilutions of the titrated MARV 371 bat strain spiked into appropriate biological specimens.

**Virus isolation and immunofluorescence assay.** Virus isolation was attempted on all MARV RNA-positive oral and urine specimens. Monolayers of Vero E6 cells (American Type Culture Collection, CRL-1586; mycoplasma-free) in 25 cm$^2$ flasks were inoculated with specimen and incubated for 1 h at 37 °C in the presence of 5% $CO_2$. Following the addition of maintenance media (DMEM containing 2% Thermo Scientific HyClone fetal bovine serum (Fisher Scientific), 100 units ml$^{-1}$ penicillin (Life Technologies), 100 µg ml$^{-1}$streptomycin (Life Technologies) and 2.50 µg ml$^{-1}$ amphotericin B (Life Technologies), cultures were incubated at 37 °C in the presence of 5% $CO_2$. Fresh maintenance media were added at 1 and 7 DPI, and the cultures were monitored through 14 DPI.

As described previously[14], all cultures were tested by immunofluorescence assay for MARV antigen at 7 and 14 DPI. Immunofluorescence assay spot slides prepared from inoculated Vero E6 cells were gamma-irradiated and then fixed in acetone. After being incubated with a 1:100 dilution of rabbit anti-MARV polyclonal (in-house) or normal rabbit serum (negative control; in-house) for 30 min at 37 °C, rinsed two times with 1 × PBS for 10 min, incubated with a 1:40 dilution of goat anti-rabbit fluorescein isothiocyanate (Capel-ICN Pharmaceuticals, Aurora, OH, USA) for 30 min at 37 °C, rinsed with 1 × PBS for 7 min, stained with Eriochrome Black T (in-house) for 7 min and rinsed with 1 × PBS for 7 min, the slides were observed under a fluorescence microscope.

**Serology.** As described previously[14], ELISA plates were coated with 50 ng per well of purified recombinant Marburg Angola NP or Reston NP expressed in *Escherichia coli* (GenScript, Piscataway, NJ, USA) diluted in PBS containing 1% thimersol. The plates were incubated overnight at 4 °C and then washed with PBS containing 0.1% Tween-20 (PBS-T). A 1:100 dilution of gamma-irradiated bat whole blood in masterplate diluent (PBS containing 5% skim milk powder, 0.5% tween-20 and 1% thimersol) was then added to the first well and fourfold serial dilutions in serum diluent (PBS containing 5% skim milk and 0.1% tween-20) were performed through 1:6,400. After incubating for 1 h at 37 °C, the plates were washed with PBS-T and bound antibody was detected using a 1:2,000 dilution of anti-goat bat IgG (Bethyl Laboratories, Montgomery, TX, USA) in serum diluent. The manufacturer product datasheet states that this antibody reacts specifically with bat IgG and with light chains common to other bat immunoglobulins. Following incubation with the secondary antibody for 1 h at 37 °C, the plates were washed twice with PBS-T and the 2-Component ABTS Peroxidase System (KPL, Gaithersburg, MD, USA) was added. The substrate was allowed to incubate for 30 min at 37 °C before reading the plates on a microplate spectrophotometer at 410 nm. The adjusted sum OD values were calculated by subtracting the ODs at each fourfold dilution wells coated with Reston NP from their corresponding wells coated with Marburg Angola NP. The average adjusted sum OD of duplicate runs was reported and the threshold for seropositivity was set at 0.95, as previously

described[14]. The mean and s.d. of the adjusted sum ODs of bats from the ERB breeding colony were used to plot a frequency distribution and calculate a value greater than the mean + 3 s.d. If a bat has an adjusted sum OD ≥ 0.95, we are > 99.7% confident that it was infected with MARV and seroconverted.

**Statistical analyses.** The number of bats used were based on the size of the transmission cage (required to fit in a bio-flow isolator unit) and the reproductive capacity of the ERB breeding colony. An effort was made to sex-match bats according to group and unit. No other randomization methods were used throughout the study. Investigators were not blinded during the study.

Differences between groups were analysed using nonparametric Mann–Whitney *U*-tests (sample distributions were not normal as determined by the D'Agostino and Pearson test), the unpaired *t*-test (sample distribution was normal as determined by the D'Agostino and Pearson test) or descriptive statistics (sample sizes were not sufficient for statistical significance testing). Mann–Whitney *U*-tests were used to determine if MARV RNA loads in blood and oral swab specimens differed significantly (two-tailed $P < 0.05$) between male and female inoculated bats (GraphPad Prism 6 software, La Jolla, CA). Mann–Whitney *U*-tests were also used to determine whether MARV RNA loads or day of collection differed significantly between MARV isolation-positive and -negative oral swabs. The unpaired *t*-test was used to determine whether peak MARV IgG antibody levels differed significantly between inoculated and contact bats.

A Lorenz curve of cumulative percentage of the inoculated bat population versus cumulative percentage of oral MARV shedding ranked in descending order was constructed and a Gini coefficient was calculated to quantify inequality in MARV oral shedding (GraphPad Prism 6 software). Lorenz curves have traditionally been used in economics to describe inequalities in wealth[56]. Over the years, their application has been extended to the study of infectious diseases. Lorenz curves have been used to: quantify heterogeneities in host–vector contact rates to show that targeted interventions can substantially reduce infection rates[36], identify the contribution of ixodid tick functional groups in tick borne encephalitis virus transmission to the yellow-necked field mouse[35] and quantify heterogeneities in viral shedding from their respective natural reservoir avian hosts[18].

**Data availability.** The authors declare that all data supporting the findings of this study are available within the article and its supplementary information files, or from the authors upon request.

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

## Acknowledgements

We thank Peter Eworonsky, Lester Slough, Eddie Jackson, Sharon Dietz and Abiola Aminu from the Centers for Disease Control and Prevention's Comparative Medicine Branch for providing care and support of the bats. This study was funded in part by DTRA Grant HDTRA-14-1-0016, Subaward S-1340-03. The findings and conclusions in this report are those of the authors and do not necessarily represent the official position of the Centers for Disease Control and Prevention.

## Author contributions

A.J.S., B.R.A., M.E.B.J. and J.S.T. conceived and designed the experiments. A.J.S., B.R.A., M.E.B.J., T.K.S., L.S.U., J.R.S., B.E.M., J.D.C.M. and J.S.T. performed the experiments. A.J.S. analysed the data. A.J.S., B.R.A., S.T.N. and J.S.T. wrote the paper.

## Additional information

**Competing financial interests:** The authors declare no competing financial interests.

