## [Peer Review File · Nature Communications]

Reviewers' comments:

Reviewer #1 (Remarks to the Author):

The manuscript entitled "Modeling Filovirus Maintenance in Nature: Experimental Horizontal Transmission of Marburg Virus between Egyptian Rousette Bats" by Schuh et al. assesses the potential for horizontal bat-to-bat transmission of Marburg virus in experimentally infected captive fruit bats. In brief, the authors exposed 12 bats to Marburg virus and co-housed these 12 bats with 24 naïve contact bats. The animals were monitored for evidence of Marburg infection for 9 months. Importantly, oral Marburg shedding was detected in 11 of the 12 experimentally infected bats and in the late phase of the study horizontal transmission of Marburg virus from inoculated to contact bats was demonstrated. Of particular interest the authors showed that two of the experimentally infected bats were "supershedders" together accounting for 44% of the total Marburg shedding. Overall, this is a very well written and important paper that provides further insight into how filoviruses are transmitted in nature. A previous paper suggested that bat-to-bat Marburg transmission may be mediated by hematophagous arthropods present in natural roosts. In the current study the authors employed bats in the absence of arthropod vectors and present data that strongly supports their view that arthropods do not play a major role in bat-to-bat transmission. The duration of the study at 9 months provided results not seen in previous studies of much shorter duration. This is no small feat in keeping bats this long in BSL-4 containment.

Minor comments:

1. The authors use TCID50 to quantitate infectious virus? As many filovirus papers use PFU what is the relationship between TCID50 and PFU?
2. Lines 58-75. It would be helpful if the strains of Marburg virus were noted here. While the various strains of Marburg may be genetically similar could there be differences in strain that could also explain differences seen between the Paweska et al. 2015 study and the authors' current study? For example, it is thought that the Angola strain while very similar to other Marburg strains is more lethal in primates?

Reviewer #2 (Remarks to the Author):

This manuscript represents a milestone in a long-running sequence of methodically planned and executed investigations to identify reservoir hosts of Marburg virus and uncover mechanisms for transmission and perpetuation of infection within host populations. The findings go a long way towards providing answers, and also suggest pathways for spillover of infection to species susceptible to disease, including humans. The prolonged period of observation of the bats on experiment appears to have been key to the success. The work is a testimony to the stamina and steadfastness of purpose of the research team.

The text is of necessity densely written, and demands careful reading, but appears to be

refreshingly free of errors. The abbreviation MARV is introduced for Marburg virus in line 26 and used throughout the text, but in lines 299 and 301 there is inconsistent use of 'marburgvirus'. The abbreviation MPI is introduced for 'months post-infection' in line 142 and used throughout the text, but needlessly explained again in line 386.

Lines 311 to 313 read: 'If the MARV-ERB relationship is reflective of the ebolavirus-bat relationship, this may explain why there is only scant serological evidence of Ebola virus infection in other bat species.' It is not clear what is implied by the word 'other' - other than what?

In line 369 there should be a space between 371 and bat strain.

Bob Swanepoel

Reviewers' comments:

Reviewer #1 (Remarks to the Author):

The manuscript entitled “Modeling Filovirus Maintenance in Nature: Experimental Horizontal Transmission of Marburg Virus between Egyptian Rousette Bats” by Schuh et al. assesses the potential for horizontal bat-to-bat transmission of Marburg virus in experimentally infected captive fruit bats. In brief, the authors exposed 12 bats to Marburg virus and co-housed these 12 bats with 24 naïve contact bats. The animals were monitored for evidence of Marburg infection for 9 months. Importantly, oral Marburg shedding was detected in 11 of the 12 experimentally infected bats and in the late phase of the study horizontal transmission of Marburg virus from inoculated to contact bats was demonstrated. Of particular interest the authors showed that two of the experimentally infected bats were “supershedders” together accounting for 44% of the total Marburg shedding. Overall, this is a very well written and important paper that provides further insight into how filoviruses are transmitted in nature. A previous paper suggested that bat-to-bat Marburg transmission may be mediated by hematophagous arthropods present in natural roosts. In the current study the authors employed bats in the absence of arthropod vectors and present data that strongly supports their view that arthropods do not play a major role in bat-to-bat transmission. The duration of the study at 9 months provided results not seen in previous studies of much shorter duration. This is no small feat in keeping bats this long in BSL-4 containment.

Minor comments:

1. The authors use TCID₅₀ to quantitate infectious virus? As many filovirus papers use PFU what is the relationship between TCID₅₀ and PFU?

Yes, TCID₅₀ assays were used to quantify infectious virus. A recent study (Smither *et al.*, *Journal of Virological Methods* 193 (2013) 565-571) compared the TCID₅₀ and plaque assay methods for measuring filovirus infectivity and found that TCID₅₀ counts, using a strain of Ebola virus (*Family Filoviridae*) carrying green fluorescent protein, were almost 10 times greater than that for plaque assay counts. The exact conversion formula reported by this study was: TCID₅₀=plaque forming units (PFU)*7.33. This formula with reference has been inserted into the “Methods” section at line 371.

2. Lines 58-75. It would be helpful if the strains of Marburg virus were noted here. While the various strains of Marburg may be genetically similar could there be differences in strain that could also explain differences seen between the Paweska et al. 2015 study and the authors' current study? For example, it is thought that the Angola strain while very similar to other Marburg strains is more lethal in primates?

The strains of Marburg virus used in the previous studies are now noted in the introduction between lines 60 and 70. We agree that it is possible host-specific genetic mutations present in the human-derived SPU 148/99/1 strain of Marburg virus used in the Paweska transmission study (versus the bat-derived bat 371 strain of MARV used in the current study) may have led to the observed low-level of viral shedding and decreased infectivity of the experimentally infected bats. A statement to this effect has now been included in the “Discussion” section between lines 249 and 252.

Reviewer #2 (Remarks to the Author):

This manuscript represents a milestone in a long-running sequence of methodically planned and executed investigations to identify reservoir hosts of Marburg virus and uncover mechanisms for transmission and perpetuation of infection within host populations. The findings go a long way towards providing answers, and also suggest pathways for spillover of infection to species susceptible to disease, including humans. The prolonged period of observation of the bats on experiment appears to have been key to the success. The work is a testimony to the stamina and steadfastness of purpose of the research team.

The text is of necessity densely written, and demands careful reading, but appears to be refreshingly free of errors. The abbreviation MARV is introduced for Marburg virus in line 26 and used throughout the text, but in lines 299 and 301 there is inconsistent use of 'marburgvirus'. The abbreviation MPI is introduced for 'months post-infection' in line 142 and used throughout the text, but needlessly explained again in line 386.

“Marburgvirus” has been replaced with “MARV” in lines 302 and 304 and “Months post infection” was replaced with “MPI” in line 388.

Lines 311 to 313 read: 'If the MARV-ERB relationship is reflective of the ebolavirus-bat

relationship, this may explain why there is only scant serological evidence of Ebola virus infection in other bat species.' It is not clear what is implied by the word 'other' - other than what?

To clarify this statement “...other bat species” has been changed to “wild-caught bats” at line 315.

In line 369 there should be a space between 371 and bat strain.

A space has been inserted between “371” and “bat” in lines 371 and 427.

Bob Swanepoel

REVIEWERS' COMMENTS:

Reviewer #1 (Remarks to the Author):

The authors have adequately addressed my comments.